# Efficacy of prophylactic tranexamic acid among parturient at increased risk for postpartum hemorrhage undergoing cesarean delivery: A systematic review and meta-analyses of randomized controlled trials

**Mohamed Aziz Daghmouri**[1]*, **Simon Crequit**[2], **Amelie Madeuf**[3], **Wael Chaabane**[4], **Ortéga Laurent**[4], **Patrick Lafforgue**[4], **Bechir Azzouzi**[4], **Ines Ouaz**[4], **Mohamed Ali Chaouch**[5]

1 Department of Anesthesia, Montreuil Intercommunal Hospital Center, Montreuil, France, 2 Department of Gynecology, Montreuil Intercommunal Hospital Center, Montreuil, France, 3 Department of Gynecology, Perpignan Hospital Center, Perpignan, France, 4 Department of Emergency, Perpignan Hospital Center, Perpignan, France, 5 Department of Visceral and Digestive Surgery, Monastir University Hospital, Monastir, Tunisia

* aziz.daghmouri@gmail.fr

## Abstract

### Background

Postpartum hemorrhage (PPH) remains a significant cause of maternal morbidity and mortality worldwide. Prophylactic pharmacological interventions, especially tranexamic acid (TXA), are under evaluation for their efficacy in preventing PPH during cesarean sections in high-risk women. This systematic review and meta-analysis aimed to assess the effectiveness of prophylactic tranexamic acid among parturients at increased risk for postpartum hemorrhage undergoing cesarean delivery.

### Methods

This systematic review and meta-analysis followed the PRISMA 2020 guidelines and was registered in the PROSPERO database (ID: CRD42024520613). We conducted a comprehensive search in several bibliographic databases for randomized controlled trials comparing prophylactic tranexamic acid to placebo in parturients at increased risk for PPH undergoing cesarean delivery published until 1st, 2024. We sought trials in the United States National Library of Medicine, Cochrane Database of Systematic Reviews (CDSR) and the Cochrane Central Register of Controlled Trials (CENTRAL), Embase, National Institutes of Health PubMed/MEDLINE, Web of Science, and Google Scholar databases. Primary outcomes included intraoperative blood loss, while secondary outcomes covered hemoglobin loss, transfusion needs, the incidence of PPH, and side effects.

**Data availability statement:** All relevant data are within the manuscript and its Supporting Information files.

**Funding:** The author(s) received no specific funding for this work.

**Competing interests:** No authors have competing interests.

## Results

Eleven randomized controlled trials involving 1627 patients were included. The meta-analysis revealed that TXA significantly reduces intraoperative blood loss compared to placebo (Mean Difference (MD) = −343.89, 95% CI [−394.34, −293.43], p < 0.00001). Furthermore, TXA was associated with lower rates of blood transfusions and PPH. The heterogeneity was substantial across studies but reduced significantly in subgroup analyses. No significant differences in side effects, hysterectomy, or additional uterotonic use were observed between the two groups.

## Conclusions

Prophylactic tranexamic acid significantly reduces intraoperative blood loss and the incidence of PPH in parturients at increased risk undergoing cesarean delivery, without increasing side effects. These findings support the broader use of TXA in this high-risk group, although further research is necessary to explore long-term outcomes and optimal administration protocols.

## Introduction

Primary postpartum hemorrhage (PPH) is defined as cumulative blood loss exceeding 500 mL following a vaginal delivery or 1000 mL following cesarean delivery, or the occurrence of hypovolemia symptoms within 24 hours post-delivery [1,2]. Despite extensive collaborative efforts at various levels, PPH remains the leading cause of maternal morbidity and mortality globally, particularly in low-income countries [3]. Numerous risk factors for PPH have been identified, including predelivery anemia, placental abnormalities, prolonged labor, preeclampsia, fetal macrosomia, amnionitis, fibroids, and instrumental vaginal delivery [4,5]. According to the latest recommendations from the International Federation of Gynecology and Obstetrics (FIGO) on managing PPH, the sole prophylactic pharmacological intervention demonstrated to reduce PPH is the immediate administration of uterotonics after delivery [3]. Antifibrinolytics, such as tranexamic acid (TXA), have been systematically employed to prevent bleeding following head injuries, orthopedic surgeries, and major cardiac and abdominal surgeries [6–9]. The efficacy of TXA in reducing blood loss is primarily attributed to its mechanism of inhibiting plasminogen activation, which subsequently prevents fibrinolysis [10].

Although TXA has demonstrated consistent efficacy in various surgical contexts, its prophylactic use during cesarean delivery remains subject to ongoing debate. This controversy is largely driven by several unresolved concerns: the limited and heterogeneous safety data, particularly regarding thromboembolic risks in pregnant women; variability in clinical outcomes across different risk groups and surgical settings; a lack of consensus on the optimal timing and dosage for administration; and a scarcity of robust evidence in high-risk obstetric populations. A Cochrane review by Novikova et al concluded, based on studies of mixed quality, that TXA reduces postpartum

blood loss and prevents PPH and blood transfusions following a vaginal birth and cesarean in women at low risk of PPH [11]. The review also emphasized the need for further investigation into the use of TXA in women at high risk of PPH. A more recent meta-analysis by Cheema et al., which included 50 randomized controlled trials (six involving only high-risk patients), concluded that TXA may reduce the risk of blood loss during cesarean deliveries, with a more pronounced benefit observed in high-risk patients [12]. However, the review included a heterogeneous patient population, limiting its applicability to high-risk groups. Given this gap, the application of TXA for the prevention of PPH in high-risk women undergoing cesarean section has been identified as a research priority [13].

To address these limitations, we conducted an updated systematic review and meta-analysis focused exclusively on high-risk parturients undergoing cesarean section. By applying more rigorous inclusion criteria and incorporating recently published randomized controlled trials, our aim was to better delineate the efficacy and safety of prophylactic TXA in this specific population. This work intends to inform clinical practice by providing targeted evidence for TXA use in a context where the balance between benefit and potential risk is especially critical.

## Methods

This systematic review and meta-analysis follows the Preferred Reporting Items for Systematic Review and Meta-analysis (PRISMA) Guidelines 2020 [14] and is checked according to the AMSTAR 2 (Assessing the methodological quality of systematic reviews) guidelines [15]. The protocol is registered in the PROSPERO database (ID: CRD42024520613).

### Bibliographic sources

We performed an electronic search of the relevant literature and limited our search to data published until March 1st, 2024. We did not use language restrictions. We sought trials in the United States National Library of Medicine, Cochrane Database of Systematic Reviews (CDSR) and the Cochrane Central Register of Controlled Trials (CENTRAL), Embase, National Institutes of Health PubMed/MEDLINE, Web of Science, and Google Scholar databases. The MEDLINE and Embase strategies were run simultaneously as a multi-file search in Ovid, and the results were de-duplicated using the Ovid duplication tool. We used the following keywords: ''Tranexamic acid," "prophylaxis", "cesarean", "delivery", "intraoperative bleeding", "postpartum hemorrhage", "high-risk pregnancy" and "randomized controlled trial". We checked the reference list of included trials manually to identify additional studies. Additionally, we searched several clinical trial registries (ClinicalTrial.gov, Current Controlled Trials, Australian New Zealand Clinical Trials Registry (www.actr.org.au), Prospero registration, and University Hospital Medical Information Network Clinical Trials Registry (www.umin.ac.jp/ctr) to identify ongoing trials.

### Study selection

Two authors performed independent and blinded record screening. Disagreements were resolved by discussion after consulting a third review team member. Then the full texts of all selected studies were screened according to predefined inclusion and exclusion criteria. Included studies were exclusively randomized controlled trials. Only articles published in peer-reviewed journals were considered. Data from non-comparative studies, review articles, editorial letters, abstracts only, comments, and case series (fewer than ten cases) were excluded. Studies were also excluded if they compared combinations of induction agents.

### Assessment of risk of bias

The Cochrane tool for bias assessment was used to assess the risk of bias in RCTs (RoB2) [16]. We evaluated the bias in five distinct domains (A. randomization process, B. deviations from intended interventions, C. the bias in the measurement of outcome, D. bias to missing outcome data, and E. bias in selecting the reported results). Within each domain, one or more signaling questions lead to judgments of "low risk of bias", "some concerns", or "high risk of bias".

## Data extraction and outcomes

Data, including the first author's name, year of publication, country, sample size, age, gestational age, high bleeding risk considered, uterotonic agent, experimental intervention, duration of surgery, outcomes, and follow-up, were extracted from the studies. We conducted our search based on the Population, Intervention, Comparator, and Outcome (PICO) approach.

**Population.** The population of interest included adult parturients (≥18 years) undergoing cesarean delivery who were considered at increased risk for primary PPH. High-risk status was defined based on pre-specified obstetric or medical conditions known to increase the likelihood of significant bleeding. These included:

- Obstetric risk factors: placenta previa, placenta accreta, multiple gestation, polyhydramnios, macrosomia, uterine fibroids.

- Intrapartum factors: prolonged labor, chorioamnionitis, prolonged oxytocin use, general anesthesia.

- Maternal conditions: anemia, high parity, or history of PPH.

These criteria were based on established clinical guidelines and previous literature identifying risk factors associated with increased bleeding during delivery [5].

**Intervention.** The intervention under investigation was prophylactic administration of TXA. Across included studies, TXA was administered intravenously, either as a single bolus dose before or after skin incision or as a bolus followed by continuous infusion, depending on the study protocol. The typical dose ranged between 10 mg/kg and 1 g IV, aligning with current clinical practice.

**Control group.** The comparator group in each study received a placebo (commonly normal saline or similar inert solution), matched in volume and administration timing to the TXA group, ensuring the blinding of participants and care providers in all randomized controlled trials included.

**Outcomes.** The primary outcome was the amount of intraoperative blood loss, which is the most immediate and quantifiable indicator of hemorrhagic risk during cesarean delivery.

Secondary outcomes included 24-hour hemoglobin concentration loss as a marker of total blood loss, the need for blood transfusion (including the number of units transfused), and the incidence of primary postpartum hemorrhage (PPH), defined as blood loss >1000 mL or RBC transfusion within 48 hours. Additional outcomes were the use of supplementary uterotonics, incidence of hysterectomy, and adverse events potentially related to tranexamic acid, such as nausea, vomiting, dizziness, or thromboembolic events. These outcomes provided a broader assessment of both the efficacy and safety of prophylactic TXA in high-risk cesarean deliveries [17]. Subgroup analyses were performed according to the methods of TXA administration, specifically comparing single-dose administration to continuous infusion. In case of unclear bias domains or missing primary outcomes information, authors were contacted by e-mail.

## Statistical analysis

We used the RevMan 5.4 statistical package from the Cochrane Collaboration for meta-analysis [18]. We selected the mean difference (MD) as an effective measure for continuous data. Odds ratios (OR) with 95% confidence intervals (95% CI) were calculated for dichotomous variables. The random-effects model was used, and the significance threshold was fixed at 0.05. When mean and standard deviation (SD) were not reported, they were estimated from the provided range (R) and median based on the formula described by Hozo et al. [19].

## Assessment of heterogeneity

To assess heterogeneity, three strategies were used:

1. The Cochrane $Chi^2$ test (Q-test), the $Tau^2$ which is the variance of true effects, and 95% predictive interval (index of dispersion) to estimate the degree of heterogeneity [20]. We calculated the predictive interval using a Comprehensive

Meta-analysis prediction interval. The values less than 25% indicated no heterogeneity, between 25% and 50% indicated moderate heterogeneity and more than 50% indicated substantial heterogeneity.

2. Graphical exploration with funnel plots [21].

3. Sensitivity analysis with a subgroup analysis when applicable [22]. Subgroup analyses were carried out, if feasible, to assess potential sources of heterogeneity.

### Summary of findings

Two authors independently assessed the evidence of the primary outcomes using Grading of Recommendations Assessment, Development, and Evaluation (GRADE) [23]. We considered the study limitations in terms of the constancy of effect, imprecision, indirectness, and publication bias. We assessed the certainty of the evidence as high, moderate, low, or very low. If appropriate, we considered the following criteria for upgrading the evidence: large effect, dose-response gradient, and plausible confounding effect. We used the methods and recommendations described in sections 8.5 and 8.7 and chapters 11 and 12 of the Cochrane Handbook for Systematic Reviews of Interventions. We used GRADEpro GDT software to prepare a summary of the findings tables. We explained the reasons for downgrading or upgrading the included studies using footnotes and comments.

## Results

### Bibliographic research

The literature search identified 218 articles, from which 20 were selected for full-text review and finally 11 studies were included [24–34]. The detailed PRISMA flow diagram is presented in Fig 1. Nine articles were excluded for the following reasons: four studies were systematic reviews or meta-analyses [35–38], two studies assessed inadequate outcomes [39,40] two studies were non-randomized controlled trials [41,42], and one study was only a protocol [43]. The demographic data of the retained studies are presented in Table 1. The risk of bias assessment using the RoB 2 was presented in Table 2. A total of 1627 patients (812 patients in the TXA group and 815 patients in the placebo group) were included in this study. These studies were published from December 2017 to February 2024. Four studies were from Egypt, two were from Nigeria, two were from India, and the others were from Singapore, France, and Saudi Arabia. Concerning methods of TXA administration, a single bolus was used in nine trials and continuous infusion was used in only two trials [27,30].

### Primary outcome: Intraoperative blood loss

All the included RCTs reported data on intraoperative blood loss. It was evaluated in 812 patients in the TXA group and 815 patients in the placebo group. Pooled results showed that intraoperative blood loss was significantly reduced in the TXA group compared to placebo but with high heterogeneity (MD = −343.89, 95% CI [−394.34, −293.43] $p < 0.00001$; $Tau^2 = 3997.65$ ($I^2 = 74\%$)) (Fig 2). Subgroup analyses demonstrated that within the single dose subgroup, TXA markedly decreased intraoperative blood loss (MD = −347.03, 95% CI [−393.72, −300.43] $p < 0.00001$). Conversely, in the continuous infusion subgroup, TXA did not significantly impact blood loss (MD = −585.6, 95% CI [−1635.60, 464.41] $p = 0.0003$). Nonetheless, the observed heterogeneity across studies remained substantial. Graphical evaluation using a funnel plot revealed that upon exclusion of the study by Neumann et al. [34], the significance of the difference between the groups persisted, and there was a reduction in heterogeneity (MD = −369.80, 95% CI [−403.82, −335.79]; $I^2 = 37\%$).

### Secondary outcomes

**24-hours hemoglobin loss.** Ten studies assessed data regarding 24-hour hemoglobin loss [24,26–34]. It was evaluated in 770 patients in the TXA group and 767 patients in the placebo group. They showed that in the TXA group

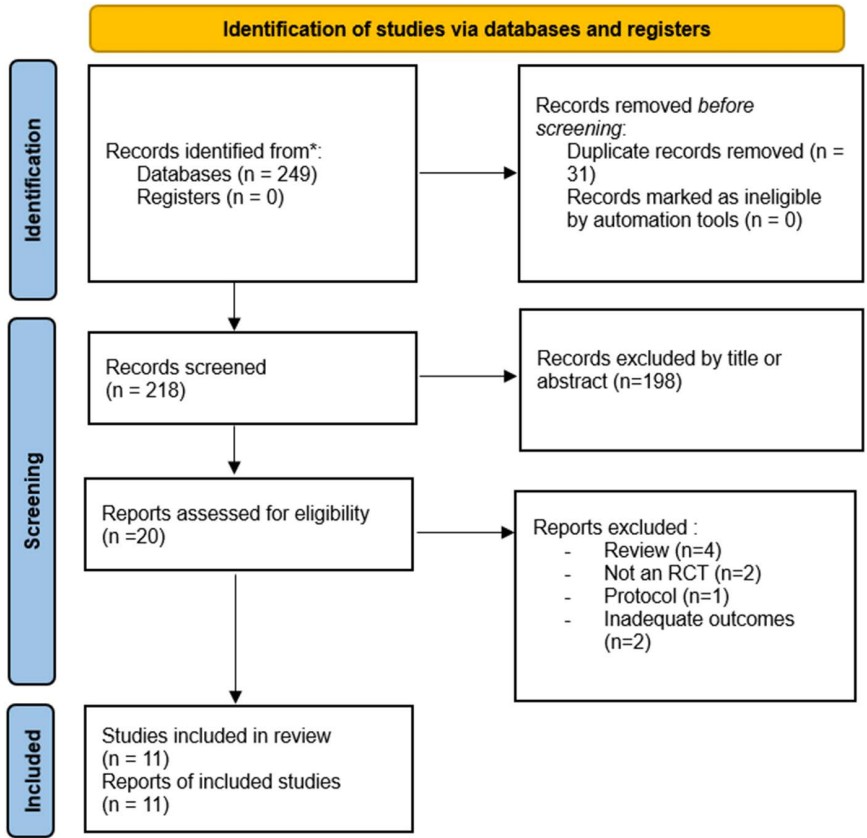

**Fig 1. PRISMA 2020 flow diagram of the retained studies.**

hemoglobin loss was significantly reduced compared with placebo (MD = −0.87, 95% CI [−1.13, −0.61]; p < 0.00001) (Fig 3A). There was a low heterogeneity among the studies (Tau$^2$ = 0.16 (I$^2$ = 99%)).

**Transfusion outcomes.** Eight studies reported data on blood transfusion [24,25,28–33]. It was reported in 41 out of 568 patients in the TXA group and 105 out of 571 patients in the placebo group. They found that 7.2% of patients were transfused in the TXA group versus 18.3% of patients in the placebo group, with a significant difference and a low heterogeneity among the studies (OR = 0.29, 95% CI [0.13, 0.61] p = 0.001; Tau$^2$ = 0.72 (I$^2$ = 65%)) (Fig 3B). Only two studies [27,30] assessed data on the number of RBCs transfused and showed that it was reduced in the TXA group compared to placebo (MD = −1.44, 95% CI [−2.03, −0.86]; I$^2$ = 0%).

**Incidence of PPH.** The incidence of PPH was assessed in six studies [26,28–31,33]. It was reported in 121 out of 530 patients in the TXA group and 207 out of 527 patients in the placebo group. They found that the TXA group was associated with a reduced significant incidence of PPH compared to the placebo group (OR = 0.30, 95% CI [0.14, 0.67]; p = 0.003) (Fig 3C). There was a low heterogeneity among the studies (Tau$^2$ = 0.64 (I$^2$ = 79%)). Subgroup analyses demonstrated that within the single dose subgroup, the TXA markedly decreased the incidence of PPH and reduced the heterogeneity between studies (OR = 0.23, 95% CI [0.13, 0.39]; I$^2$ = 31%).

**Additional uterotonic use.** Eight studies reported data on the need for additional uterotonic [24,28–34]. It was assessed in 99 out of 632 patients in the TXA group and 217 out of 629 patients in the placebo group. Pooled results found that additional uterotonic was needed in 15.7% of patients in the TXA group versus 34.5% in the placebo group (OR = 0.26, 95% CI [0.14, 0.48]; p < 0.0001) (Fig 3D). There was a low heterogeneity among the studies (Tau$^2$ = 0.49 (I$^2$ = 71%))

**Table 1. Demographic data of the retained studies.**

| Authors (year) | Country | Sample size | Age (years) | Gestational age (weeks) | High bleeding risk considered | Uterotonic agents | Experimental intervention | Duration of surgery | Outcomes | Follow-up |
|---|---|---|---|---|---|---|---|---|---|---|
| 1-Abbas et al (2019) | Egypt | 62 patients (31 vs 31) | 30.65±2.65 | 36.55±0.7 | Placenta previa (bilateral uterine artery ligation) | oxytocin | IV 1g TXA before skin incision | 3.9±0.9 vs 4.9±1.5 | - Total estimated blood loss <br> - Hb concentration | 24 hours |
| 2-Lee et al (2023) | Singapore | 90 patients (42 vs 48) | 33.75±4.3 | 38.55±0.75 | History of PPH, Anemia, macrosomia, placenta previa | Oxytocin | IV 1g TXA 10min before skin incision | 52±18.7 vs 53.4±22.4 | - estimated blood loss <br> - side effects | 48 hours |
| 3-Dawoud et al (2023) | Egypt | 300 patients (115 vs 115) | 29.75±4.65 | 38.52±0.96 | Anemia, PE, macrosomia, History of PPH | Oxytocin | IV 1g TXA 15min before skin incision | 73.88±14.95 vs 74.24±15.26 | - Intraoperative blood loss <br> - incidence of PPH <br> - Hb concentration | 48 hours |
| 4-Ibrahim et al (2019) | Saudi Arabia | 46 patients (23 vs 23) | 31.45±5.45 | – | Placenta accreta | – | IV 10mg/kg TXA after cord clamping and continued infusion 10mg/kg/h | – | - Intraoperative blood loss <br> - blood transfusion | 24 hours |
| 5-Ifunanya et al (2019) | Nigeria | 168 patients (84 vs 84) | 28.4±5.3 | 38±1.4 | Hypertensive disorders, chorioamnionitis, placenta previa, History of PPH, macrosomia | Oxytocin | IV 1g TXA 10min before skin incision | – | - Additional uterotonic <br> - incidence of PPH <br> - Estimated blood loss | 48 hours |
| 6-Ortuanya et al (2024) | Nigeria | 200 patients (100 vs 100) | 31.92±4.57 | – | History of PPH, uterine fibrinoid, placenta previa, severe PE | Oxytocin | IV 1g TXA 10min before skin incision | 56.66±15.67 vs 59.95±17.14 | - Intraoperative blood loss <br> - Hematocrit change <br> - blood transfusion | 48 hours |
| 7-Sentilhes et al (2022) | France | 319 patients (160 vs 159) | 33.9±4.9 | – | Multiple pregnancies | Oxytocin or carbetocin | IV 1g TXA after birth followed by 2h continued infusion | – | - Intraoperative blood loss <br> - Transfusion | 48 hours |
| 8-Shady et al (2017) | Egypt | 80 patients (40 vs 40) | 29.55±2.55 | 36.41±0.88 | Placenta previa | Oxytocin | IV 1g TXA before skin incision | 48.05±5.49 vs 48.13±5.88 | - Blood loss postoperatively | – |
| 9-Shalaby et al (2022) | Egypt | 160 patients (80 vs 80) | 28.7±4.65 | 38.0±1.1 | Overdistended uterus, placenta previa, anemia, history of PPH | Oxytocin | IV 1g TXA 15min before skin incision | 49.9±19.7 vs 47.8±19.1 | - Intraoperative blood loss | 48 hours |
| 10-Sujata et al (2016) | India | 60 patients (31 vs 29) | 29.83±4.23 | – | Chorioamnionitis, use of oxytocin, placenta previa, polyhydramnios | Oxytocin | IV 10mg/kg TXA 10min before skin incision | – | - Additional uterotonic | 48 hours |
| 11-Neumann et al (2024) | India | 212 patients (106 vs 106) | 25.8±4.1 | – | Obesity, multiple pregnancy, abnormally implanted placenta, polyhydramnios, and macrosomia | Oxytocin | IV 1g TXA 10min before skin incision | – | - Intraoperative blood loss <br> - side effects | 48 hours |

IV: intravenous; TXA: tranexamic acid; Hb: hemoglobin; PPH: primary postpartum hemorrhage; PE: pre-eclampsia.

**Table 2. Risk of bias 2 assessment of the included studies.**

| Authors | Randomization process | Deviations from intended interventions | Bias in measurement of outcome | Bias to missing outcome data | Bias in selecting the reported results | Overall bias |
|---|---|---|---|---|---|---|
| Abbas et al. | Low risk | Low risk | Low risk | Low risk | Low risk | Low risk |
| Abdel-Rasheed et al. | Low risk | Some concerns | Some concerns | ow risk | Low risk | Some concerns |
| Hamed et al. | Low risk | Low risk | Low risk | Some concerns | Low risk | Some concerns |
| Ifunanya et al. | Low risk | Low risk | Low risk | Low risk | Low risk | Low risk |
| Lee et al. | Low risk | Low risk | Low risk | Some concerns | Low risk | Some concerns |
| Neumann et al. | Low risk | Low risk | Low risk | Low risk | Low risk | Low risk |
| Ortuanya et al. | Low risk | Low risk | Low risk | Low risk | Low risk | Low risk |
| Sentilhes et al. | Low risk | Low risk | Low risk | Low risk | Low risk | Low risk |
| Shady et al. | Low risk | High risk | Low risk | Low risk | Low risk | High risk |
| Shalaby et al. | Low risk | Low risk | Low risk | Low risk | Low risk | Low risk |
| Sujata et al. | Low risk | Low risk | Low risk | Low risk | Low risk | Low risk |

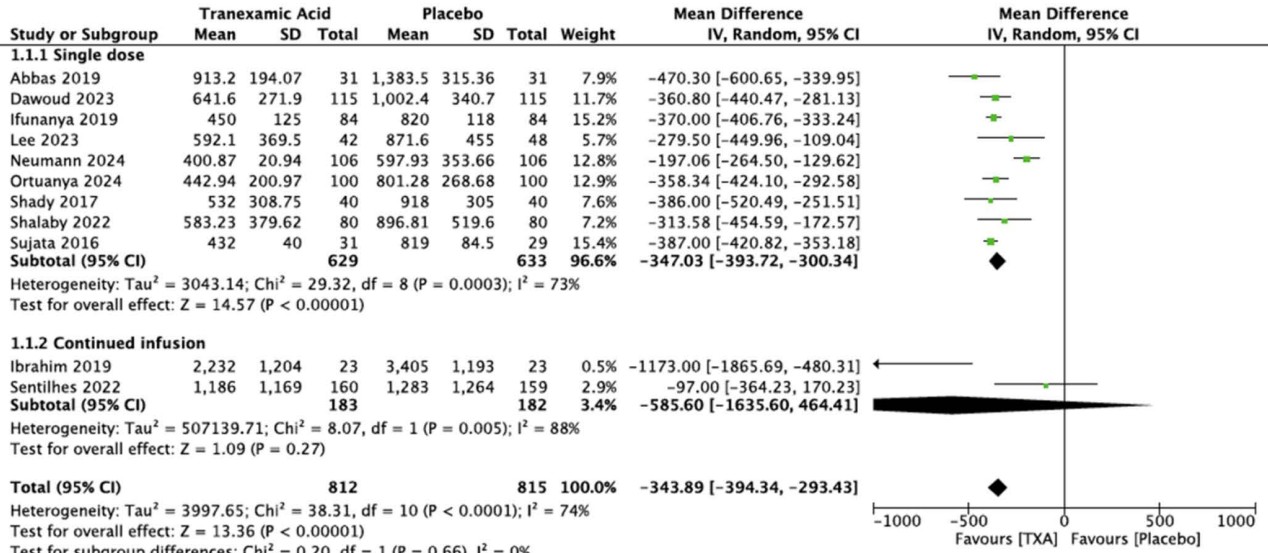

**Fig 2. Forest plot of intraoperative blood loss.**

**Hysterectomy.** Four studies assessed data on the need for a hysterectomy [24,25,29,30] and showed no significant difference between both groups (OR = 1.66, 95% CI [0.45, 6.09]; $I^2 = 17\%$).

**Side effects.** Five studies reported data on side effects. Pooled results found no significant difference between both groups (OR = 2.16, 95% CI [0.90, 5.18]; $I^2 = 32\%$).

**Reporting of the effects of TXA.** A summary of the evidence is presented in Table 3. Compared to placebo, TXA was associated with a substantial reduction in intraoperative blood loss, particularly in studies using a single-dose administration. The use of TXA was also likely to reduce the incidence of postpartum hemorrhage, the need for additional uterotonics, and the rate of blood transfusions, with a corresponding trend toward higher hemoglobin concentrations at 24 hours post-delivery—again more evident in the single-dose subgroup.

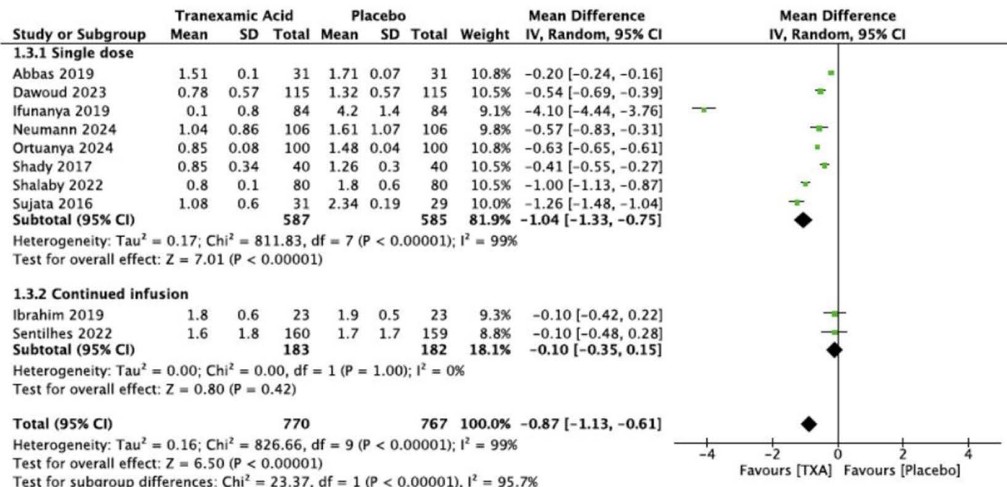

**A. 24-Hemoglobin loss**

| Study or Subgroup | Tranexamic Acid Mean | SD | Total | Placebo Mean | SD | Total | Weight | Mean Difference IV, Random, 95% CI |
|---|---|---|---|---|---|---|---|---|
| **1.3.1 Single dose** | | | | | | | | |
| Abbas 2019 | 1.51 | 0.1 | 31 | 1.71 | 0.07 | 31 | 10.8% | -0.20 [-0.24, -0.16] |
| Dawoud 2023 | 0.78 | 0.57 | 115 | 1.32 | 0.57 | 115 | 10.5% | -0.54 [-0.69, -0.39] |
| Ifunanya 2019 | 0.1 | 0.8 | 84 | 4.2 | 1.4 | 84 | 9.1% | -4.10 [-4.44, -3.76] |
| Neumann 2024 | 1.04 | 0.86 | 106 | 1.61 | 1.07 | 106 | 9.8% | -0.57 [-0.83, -0.31] |
| Ortuanya 2024 | 0.85 | 0.08 | 100 | 1.48 | 0.04 | 100 | 10.8% | -0.63 [-0.65, -0.61] |
| Shady 2017 | 0.85 | 0.34 | 40 | 1.26 | 0.3 | 40 | 10.5% | -0.41 [-0.55, -0.27] |
| Shalaby 2022 | 0.8 | 0.1 | 80 | 1.8 | 0.6 | 80 | 10.5% | -1.00 [-1.13, -0.87] |
| Sujata 2016 | 1.08 | 0.6 | 31 | 2.34 | 0.19 | 29 | 10.0% | -1.26 [-1.48, -1.04] |
| Subtotal (95% CI) | | | 587 | | | 585 | 81.9% | -1.04 [-1.33, -0.75] |

Heterogeneity: Tau² = 0.17; Chi² = 811.83, df = 7 (P < 0.00001); I² = 99%
Test for overall effect: Z = 7.01 (P < 0.00001)

| Study or Subgroup | Tranexamic Acid Mean | SD | Total | Placebo Mean | SD | Total | Weight | Mean Difference IV, Random, 95% CI |
|---|---|---|---|---|---|---|---|---|
| **1.3.2 Continued infusion** | | | | | | | | |
| Ibrahim 2019 | 1.8 | 0.6 | 23 | 1.9 | 0.5 | 23 | 9.3% | -0.10 [-0.42, 0.22] |
| Sentilhes 2022 | 1.6 | 1.8 | 160 | 1.7 | 1.7 | 159 | 8.8% | -0.10 [-0.48, 0.28] |
| Subtotal (95% CI) | | | 183 | | | 182 | 18.1% | -0.10 [-0.35, 0.15] |

Heterogeneity: Tau² = 0.00; Chi² = 0.00, df = 1 (P = 1.00); I² = 0%
Test for overall effect: Z = 0.80 (P = 0.42)

Total (95% CI) — 770 / 767 — 100.0% — -0.87 [-1.13, -0.61]
Heterogeneity: Tau² = 0.16; Chi² = 826.66, df = 9 (P < 0.00001); I² = 99%
Test for overall effect: Z = 6.50 (P < 0.00001)
Test for subgroup differences: Chi² = 23.37, df = 1 (P < 0.00001), I² = 95.7%

Favours [TXA]  Favours [Placebo]

**B. Blood transfusion**

| Study or Subgroup | Tranexamic Acid Events | Total | Placebo Events | Total | Weight | Odds Ratio M-H, Random, 95% CI |
|---|---|---|---|---|---|---|
| Abbas 2019 | 4 | 31 | 17 | 31 | 13.1% | 0.12 [0.03, 0.43] |
| Ifunanya 2019 | 5 | 84 | 12 | 84 | 14.5% | 0.38 [0.13, 1.13] |
| Lee 2023 | 2 | 42 | 6 | 48 | 10.4% | 0.35 [0.07, 1.84] |
| Ortuanya 2024 | 8 | 100 | 26 | 100 | 16.4% | 0.25 [0.11, 0.58] |
| Sentilhes 2022 | 13 | 160 | 8 | 159 | 15.9% | 1.67 [0.67, 4.14] |
| Shady 2017 | 7 | 40 | 27 | 40 | 14.8% | 0.10 [0.04, 0.29] |
| Shalaby 2022 | 1 | 80 | 5 | 80 | 7.7% | 0.19 [0.02, 1.66] |
| Sujata 2016 | 1 | 31 | 4 | 29 | 7.3% | 0.21 [0.02, 1.99] |
| Total (95% CI) | | 568 | | 571 | 100.0% | 0.29 [0.13, 0.61] |
| Total events | 41 | | 105 | | | |

Heterogeneity: Tau² = 0.72; Chi² = 20.26, df = 7 (P = 0.005); I² = 65%
Test for overall effect: Z = 3.24 (P = 0.001)

Favours [TXA]  Favours [Placebo]

**C. Incidence of primary postpartum haemorrhage**

| Study or Subgroup | Tranexamic Acid Events | Total | Placebo Events | Total | Weight | Odds Ratio M-H, Random, 95% CI |
|---|---|---|---|---|---|---|
| Dawoud 2023 | 2 | 115 | 3 | 115 | 10.8% | 0.66 [0.11, 4.03] |
| Ifunanya 2019 | 10 | 84 | 42 | 84 | 20.2% | 0.14 [0.06, 0.30] |
| Ortuanya 2024 | 40 | 100 | 67 | 100 | 22.3% | 0.33 [0.18, 0.59] |
| Sentilhes 2022 | 62 | 160 | 67 | 159 | 23.4% | 0.87 [0.56, 1.36] |
| Shady 2017 | 7 | 40 | 21 | 40 | 17.7% | 0.19 [0.07, 0.53] |
| Sujata 2016 | 0 | 31 | 7 | 29 | 5.7% | 0.05 [0.00, 0.88] |
| Total (95% CI) | | 530 | | 527 | 100.0% | 0.30 [0.14, 0.67] |
| Total events | 121 | | 207 | | | |

Heterogeneity: Tau² = 0.64; Chi² = 23.52, df = 5 (P = 0.0003); I² = 79%
Test for overall effect: Z = 2.97 (P = 0.003)

Favours [TXA]  Favours [Placebo]

**D. Need of additional uterotonics**

| Study or Subgroup | Tranexamic Acid Events | Total | Placebo Events | Total | Weight | Odds Ratio M-H, Random, 95% CI |
|---|---|---|---|---|---|---|
| Abbas 2019 | 1 | 31 | 9 | 31 | 5.8% | 0.08 [0.01, 0.69] |
| Ifunanya 2019 | 6 | 84 | 28 | 84 | 13.5% | 0.15 [0.06, 0.40] |
| Neumann 2024 | 8 | 106 | 17 | 106 | 14.0% | 0.43 [0.18, 1.04] |
| Ortuanya 2024 | 39 | 100 | 68 | 100 | 16.9% | 0.30 [0.17, 0.54] |
| Sentilhes 2022 | 26 | 160 | 27 | 159 | 16.8% | 0.95 [0.53, 1.71] |
| Shady 2017 | 7 | 40 | 27 | 40 | 12.5% | 0.10 [0.04, 0.29] |
| Shalaby 2022 | 11 | 80 | 37 | 80 | 15.1% | 0.19 [0.09, 0.40] |
| Sujata 2016 | 1 | 31 | 4 | 29 | 5.4% | 0.21 [0.02, 1.99] |
| Total (95% CI) | | 632 | | 629 | 100.0% | 0.26 [0.14, 0.48] |
| Total events | 99 | | 217 | | | |

Heterogeneity: Tau² = 0.49; Chi² = 24.00, df = 7 (P = 0.001); I² = 71%
Test for overall effect: Z = 4.35 (P < 0.0001)

Favours [TXA]  Favours [Placebo]

**Fig 3. Forest plots of secondary outcomes: A. 24-hemoglobin loss, B. Blood transfusion, C. Incidence of primary postpartum haemorrhage, D. Need for additional uterotonics.**

**Table 3. Summary of findings table.**

| Outcomes | № of participants (studies) | Certainty of the evidence (GRADE) | Relative effect (95% CI) | Anticipated absolute effects | |
|---|---|---|---|---|---|
| | | | | Risk with placebo | Risk difference with Outcomes |
| Intra-op blood loss | 1627 (11 RCTs) | ⊕⊕⊕⊕ High[a,b] | – | – | MD **343.89 lower** (394.34 lower to 293.43 lower) |
| Incidence of postpartum haemorrhage | 1057 (6 RCTs) | ⊕⊕⊕○ Moderate[a,b] | OR 0.30 (0.14 to 0.67) | 393 per 1 000 | **230 fewer per 1 000** (310 fewer to 90 fewer) |
| Haemoglobin concentration | 1537 (10 RCTs) | ⊕⊕⊕○ Moderate[a,b] | – | – | MD **0.87 lower** (1.13 lower to 0.61 lower) |
| Additional uterotonic | 1261 (8 RCTs) | ⊕⊕⊕○ Moderate[a,b] | OR 0.26 (0.14 to 0.48) | 345 per 1 000 | **225 fewer per 1 000** (276 fewer to 143 fewer) |
| Blood transfusion | 1139 (8 RCTs) | ⊕⊕⊕○ Moderate[a,b] | OR 0.29 (0.13 to 0.61) | 184 per 1 000 | **123 fewer per 1 000** (155 fewer to 63 fewer) |
| Hysterectomy | 671 (4 RCTs) | ⊕⊕○○ Low[a,b] | OR 1.66 (0.45 to 6.09) | 18 per 1 000 | **11 more per 1 000** (10 fewer to 81 more) |
| Amount of transfusion | 365 (2 RCTs) | ⊕⊕○○ Low[a,b] | – | – | MD **1.44 lower** (2.03 lower to 0.86 lower) |
| Side effects | 901 (5 RCTs) | ⊕⊕○○ Low[a,b] | OR 2.16 (0.90 to 5.18) | 159 per 1 000 | **131 more per 1 000** (14 fewer to 336 more) |

*__The risk in the intervention group__ (and its 95% confidence interval) is based on the assumed risk in the comparison group and the **relative effect** of the intervention (and its 95% CI). **CI:** confidence interval; **MD:** mean difference; **OR:** odds ratio

**GRADE Working Group grades of evidence**

**High certainty:** we are very confident that the true effect lies close to that of the estimate of the effect. **Moderate certainty:** we are moderately confident in the effect estimate: the true effect is likely to be close to the estimate of the effect, but there is a possibility that it is substantially different. **Low certainty:** our confidence in the effect estimate is limited: the true effect may be substantially different from the estimate of the effect. **Very low certainty:** we have very little confidence in the effect estimate: the true effect is likely to be substantially different from the estimate of effect.

Explanations: a. Existence of a heterogeneity, b. small sample size.

However, the evidence regarding other outcomes remains uncertain. Specifically, no clear conclusions could be drawn about TXA's impact on the need for hysterectomy, the total number of red blood cell units transfused, or the incidence of adverse events, as the data for these outcomes were limited and associated with substantial variability.

## Discussion

This systematic review and meta-analysis revealed significant reductions in mean intraoperative blood loss, 24-hour hemoglobin loss, the requirement for blood transfusions, the incidence of primary postpartum hemorrhage, and the need for additional uterotonic agents among parturient at increased risk for PPH undergoing cesarean delivery who received prophylactic tranexamic acid, compared to those who received placebo. However, the two groups had no significant difference in the incidence of side effects.

Our results align with those of Cheema et al., who in their recently updated systematic review and meta-analysis of RCTs, reported that tranexamic acid may reduce intraoperative blood loss during cesarean deliveries [12]. Their sub-group analyses indicated a more pronounced benefit in high-risk patients. However, despite the high quality of evidence in this population, the conclusions remain tentative due to the limited scope, involving only six small RCTs. In our meta-analysis, a larger sample size was incorporated, encompassing 11 randomized controlled trials with a total of 1627 patients. We observed that intraoperative blood loss was significantly reduced in the group receiving tranexamic acid (TXA), demonstrating clinically meaningful benefits. Although significant findings were noted, there was moderate to high heterogeneity among the studies included in our analysis. This variability is likely attributable to the differing

methodologies employed to calculate blood loss. It is important to note that no gold standard for measuring blood loss exists, and the most objective method currently available is the formula based on changes in hemoglobin and hematocrit levels [34]. To further substantiate our findings, additional outcomes, including 24-hour hemoglobin loss and the necessity for blood transfusions, were evaluated. The pooled results reinforced the efficacy of prophylactic tranexamic acid TXA in reducing intraoperative blood loss. Moreover, the primary objective of intrapartum intervention was to reduce the incidence of PPH, especially regarding the high-risk parturient. Our results indicated that the incidence of PPH was significantly lower in the TXA group compared to the placebo group (22.8% vs. 39.4%). These findings align with those of a recent meta-analysis by Franchini et al., which involved 18 randomized control trials including 786 parturient with low risk and concluded that TXA use, compared to controls, significantly reduced the incidence of PPH, with a risk ratio of 0.40 (95% CI 0.24–0.65) [44]. For these reasons, our findings prompt a reconsideration of the most recent guidelines issued by the Royal College of Obstetricians and Gynaecologists [45]. These guidelines currently advise obstetricians to 'consider the use of intravenous tranexamic acid (IV TXA) (0.5-1.0 g), alongside oxytocin, after cesarean section to reduce blood loss in women at increased risk of postpartum hemorrhage.' Our results suggest that the efficacy of IV TXA in this context warrants further examination.

Regarding adverse events, our study observed no significant differences in side effects between the groups receiving TXA and placebos, with no major side effects reported. These findings are consistent with several previous studies assessing the efficacy of prophylactic TXA in low-risk women undergoing cesarean or vaginal delivery [44,46,47]. However, while these studies primarily reported non-thromboembolic adverse events, the risk of maternal thromboembolic complications remains a concern, especially in populations at higher baseline risk. Although current evidence does not show a significant increase in thromboembolic events with prophylactic use of TXA in obstetric populations, the available data are limited and underpowered to detect rare but serious outcomes [48]. In addition, the impact of TXA on neonatal outcomes remains insufficiently explored. While TXA crosses the placenta, available studies have not demonstrated clear evidence of neonatal toxicity or adverse effects [49,50]; however, further investigation is needed to assess any potential long-term implications. Overall, these uncertainties highlight the need for larger, adequately powered trials with extended follow-up to better define the safety profile of TXA for both mother and neonate. Nevertheless Given its low cost, ease of administration, and favorable safety profile, tranexamic acid represents a particularly attractive intervention in low-resource settings, where access to blood products and advanced obstetric care may be limited [51]. Its use could play a critical role in reducing maternal morbidity and mortality associated with postpartum hemorrhage in these regions.

Our systematic review and meta-analysis specifically focused on high-risk women undergoing cesarean delivery, a group often omitted or only addressed in sub-group analyses in previous studies. However, this study faced significant limitations, particularly concerning its small sample size, as it included only 11 small, randomized controlled trials. Additional limitations stemmed from high heterogeneity among the included studies, which was largely due to varying methods used to assess blood loss, different regimens of tranexamic acid administration (single dose versus continuous), and diverse criteria for defining high-risk status. Heterogeneity can also come from the different obstetrical settings in which TXA was tested. Furthermore, we were not able to perform a sensitivity analysis due to the unavailability of sufficient data, which may limit the robustness of some of our conclusions. When interpreting the results of this meta-analysis, the methodological quality of the studies included was considered through a formal risk of bias assessment. Although most studies were judged to be at a low risk of bias, several had domains with "some concerns," particularly concerning deviations from intended interventions and outcome reporting. These limitations may affect the internal validity of the findings and warrant cautious interpretation. Accordingly, the certainty of the evidence was evaluated using the GRADE approach, and outcomes were downgraded where risk of bias was present. Even if all patients were at high risk of PPH, we know that emergency c-sections, especially in the second phase of labor, are more likely to end up with PPH. It is the same for twin pregnancies or history of more than 2 c section Techniques were employed to mitigate this heterogeneity. Moreover, the absence of post-discharge follow-up constrained our ability to assess long-term safety for mothers and neonates.

## Conclusions

Prophylactic tranexamic acid appears to be a promising intervention for reducing intraoperative blood loss and the incidence of PPH in high-risk women undergoing cesarean deliveries. Our findings indicate that TXA is safe, well-tolerated, and cost-effective, with no significant increase in adverse effects. However, the current body of evidence remains limited in quality, necessitating further high-quality, large-scale randomized controlled trials to strengthen these conclusions. Future research should focus on standardizing dosing regimens, identifying specific subpopulations that may benefit most from TXA, and exploring long-term maternal and neonatal outcomes. Additionally, studies evaluating TXA's impact in resource-limited settings, where access to blood transfusion and surgical interventions is constrained, would provide critical insights into its global applicability. Based on the available data, we advocate for the prophylactic administration of TXA in all women at high risk of bleeding during cesarean deliveries while emphasizing the need for continued research to refine guidelines and optimize patient outcomes.

## Supporting information

**S1 File. PRISMA checklist.**
(DOCX)

## Author contributions

**Conceptualization:** Mohamed Aziz Daghmouri, Mohamed Ali Chaouch.

**Formal analysis:** Mohamed Ali Chaouch.

**Methodology:** Mohamed Aziz Daghmouri.

**Project administration:** Mohamed Aziz Daghmouri.

**Validation:** Mohamed Aziz Daghmouri, Simon Crequit, Amelie Madeuf, Wael Chaabane, Ortéga Laurent, Patrick Lafforgue, Bechir Azzouzi, Ines Ouaz, Mohamed Ali Chaouch.

**Writing – original draft:** Mohamed Aziz Daghmouri, Mohamed Ali Chaouch.

**Writing – review & editing:** Simon Crequit, Amelie Madeuf, Wael Chaabane, Ortéga Laurent, Patrick Lafforgue, Bechir Azzouzi, Ines Ouaz.

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
