## [Decision Letter · Decision Letter 0]

19 Feb 2025

Dear Dr. Daghmouri,

Thank you for submitting your manuscript to PLOS ONE. After careful consideration, we feel that it has merit but does not fully meet PLOS ONE’s publication criteria as it currently stands. Therefore, we invite you to submit a revised version of the manuscript that addresses the points raised during the review process.

We look forward to receiving your revised manuscript.

Kind regards,

Mahmoud A. Ebada, MD

Academic Editor

PLOS ONE

2. As required by our policy on Data Availability, please ensure your manuscript or supplementary information includes the following:

Reviewers' comments:

Reviewer's Responses to Questions

**Comments to the Author**

1. Is the manuscript technically sound, and do the data support the conclusions?

Reviewer #1: Yes

Reviewer #2: Yes

Reviewer #3: Yes

Reviewer #4: Yes

2. Has the statistical analysis been performed appropriately and rigorously?

Reviewer #1: Yes

Reviewer #2: Yes

Reviewer #3: Yes

Reviewer #4: Yes

3. Have the authors made all data underlying the findings in their manuscript fully available?

Reviewer #1: Yes

Reviewer #2: Yes

Reviewer #3: Yes

Reviewer #4: Yes

4. Is the manuscript presented in an intelligible fashion and written in standard English?

Reviewer #1: Yes

Reviewer #2: Yes

Reviewer #3: Yes

Reviewer #4: Yes

Reviewer #1: Hello

This is a good article

Please review the article once again in terms of writing and also improve the quality of tables and images.

Make the conclusion section more complete and provide practical and realistic suggestions for future studies based on the study findings so that researchers can use your study to conduct more quality and targeted studies in future research.

Good luck

Reviewer #2: I have identified several areas that require further clarification and improvement to enhance the manuscript’s clarity and adherence to systematic review reporting standards. My detailed comments and suggestions are outlined below.

[Abstract]

1. Mention the databases searched and time frame.

2. Add PROSPERO registration number

[Introduction]

3. It’s unclear why another sysrev is needed despite existing meta-analysis. Please explain what this study adds-whether it includes new RCTs, focuses solely on high risk patients, or uses a stricter method.

[Result]

4. Add a discussion on how the risk of bias was considered when interpreting results.

[Discussion]

5. If sensitivity analysis was not performed, acknowledge this as a limitation.

6. Discuss maternal thromboembolic risks and potential neonatal effects, citing available literature.

7. Add a brief section discussing TXA’s affordability and relevance in low-resource settings.

[Conclusion]

8. The phrase “despite the low quality of evidence” is vague and contradicts the advocacy for TXA use. Clearly state the confidence level of the findings (e.g., moderate to high quality for some outcomes but limited safety data).

9. Provide spesific recommendations for future research, such as safety in neonates, etc

Reviewer #3: The authors have adhered to the PRISMA 2020 guidelines, ensuring transparency and reproducibility in their review process. The registration of the review in the PROSPERO database further strengthens the credibility of the methodology. The comprehensive search strategy across multiple bibliographic databases and trial registries enhances the thoroughness of the literature review.

Reviewer #4: The study titled “Efficacy of Prophylactic Tranexamic Acid Among Parturient at Increased Risk for Postpartum Hemorrhage Undergoing Cesarean Delivery: A Systematic Review and Meta-Analysis of Randomized Controlled Trials” by Daghmouri et al. addresses a very important and relevant topic. However, to enhance the quality of the study, the authors should consider incorporating the following revisions:

1. This study explores an important topic; however, I do not believe this study contributes any novel findings to the existing literature. A comprehensive meta-analysis by Cheema et al., titled "Tranexamic Acid for the Prevention of Blood Loss After Cesarean Section: An Updated Systematic Review and Meta-Analysis of Randomized Controlled Trials," has already examined this subject using data from 50 studies with 6 in high risk group, whereas the present study includes 11. Additionally, the outcomes assessed here appear to have been covered in that meta-analysis, raising concerns about redundancy. Furthermore, the introduction does not highlight what novel insights this study aims to provide or how it adds to current knowledge. I encourage the authors to revise and refine their study so that it meaningfully enhances the existing literature.

2. Page 9, Introduction Section: The introduction states that tranexamic acid (TXA) has well-established efficacy in reducing blood loss across various surgical contexts but describes its use in cesarean delivery as "controversial." However, the authors do not elaborate on what specific concerns contribute to this controversy. Is it related to safety, inconsistent efficacy, optimal dosage, or patient selection? Providing a clearer explanation of the factors that make TXA's use in cesarean delivery contentious would strengthen the introduction and help contextualize the study's rationale. I encourage the authors to explicitly outline these concerns to better justify the need for this study.

3. Page 9, Introduction Section: The structure of the introduction could be improved for better readability and clarity. Currently, it consists of a single, lengthy paragraph followed by a very brief two-line paragraph. The final paragraph does not effectively highlight the study's specific aims or expected outcomes. I recommend restructuring the introduction by breaking up the long paragraph into distinct sections that logically present background information, existing research gaps, and the study’s objectives. Additionally, the study’s aims and expected contributions should be clearly articulated to provide readers with a better understanding of its purpose and significance.

4. Pages 10-12, Methods Section: The authors should explicitly state their inclusion and exclusion criteria in more detail. While they mention selecting only peer-reviewed RCTs, they need to be more specific about the parameters used for study selection. It would be helpful to clearly outline the specific patient population, intervention, control, and outcomes that guided their selection process. Currently, the methods section lacks clarity on the exact characteristics they were looking for in the included studies, which is essential for ensuring transparency, replicability, and a well-defined research scope

5. Page 12, Results Section: The authors should present their results in paragraph form rather than using bullet points, as the current format appears informal. Additionally, similar formatting should be revised in other sections of the manuscript, such as Assessment of Heterogeneity, to maintain consistency and clarity throughout the text.

**Do you want your identity to be public for this peer review?** For information about this choice, including consent withdrawal, please see our Privacy Policy

Reviewer #1: No

Reviewer #2: No

Reviewer #3: **Yes: ** Tabeer Tanwir Awan

Reviewer #4: No

---

## [Author Response · Author response to Decision Letter 1]

4 Apr 2025

Paris, April 2nd, 2025

Dear Mahmoud A. Ebada, Academic Editor of The PLOS ONE Journal,

Firstly, we gratefully thank the reviewers for the carefully and rigorous reviewing of our work, which will undoubtedly improve the quality of our manuscript. As requested, please find attached the new version of the manuscript entitled: “Efficacy of prophylactic tranexamic acid among parturient at increased risk for postpartum hemorrhage undergoing cesarean delivery: a systematic review and meta-analyses of randomized controlled trials

”.

All criticisms raised by the reviewers have been adequately addressed and the respective changes have been highlighted in yellow-colored text in the revised manuscript.

Sincerely,

Dr Mohamed Aziz DAGHMOURI, on behalf of all co-authors.

Reviewer #1

Reviewer: This is a good article. Please review the article once again in terms of writing and also improve the quality of tables and images. Make the conclusion section more complete and provide practical and realistic suggestions for future studies based on the study findings so that researchers can use your study to conduct more quality and targeted studies in future research. Good luck

• Response: We sincerely appreciate the reviewer’s positive feedback and constructive suggestions. We have carefully reviewed the manuscript to improve the writing, as well as enhanced the quality of the tables and images for better clarity. Regarding the conclusion, we believe our recommendation to use TXA in all women at high risk of bleeding during cesarean deliveries is practical and clinically relevant. However, we will further refine this section to provide more comprehensive and actionable suggestions for future research, ensuring that our study serves as a valuable foundation for future investigations. Thank you for your insightful comments and support.

• « Prophylactic tranexamic acid appears to be a promising intervention for reducing intraoperative blood loss and the incidence of PPH in high-risk women undergoing cesarean deliveries. Our findings indicate that TXA is safe, well-tolerated, and cost-effective, with no significant increase in adverse effects. However, the current body of evidence remains limited in quality, necessitating further high-quality, large-scale randomized controlled trials to strengthen these conclusions. Future research should focus on standardizing dosing regimens, identifying specific subpopulations that may benefit most from TXA, and exploring long-term maternal and neonatal outcomes. Additionally, studies evaluating TXA’s impact in resource-limited settings, where access to blood transfusion and surgical interventions is constrained, would provide critical insights into its global applicability. Based on the available data, we advocate for the prophylactic administration of TXA in all women at high risk of bleeding during cesarean deliveries while emphasizing the need for continued research to refine guidelines and optimize patient outcomes. » (p12 L312 – 322)

Reviewer #2

Reviewer: Mention the databases searched and time frame.

• Response: We sincerely appreciate the reviewer’s comment. As requested, we have now specified the databases searched and the time frame in the revised manuscript to enhance clarity and transparency.

• “…published until 1st, 2024. We sought trials in the United States National Library of Medicine, Cochrane Database of Systematic Reviews (CDSR) and the Cochrane Central Register of Controlled Trials (CENTRAL), Embase, National Institutes of Health PubMed/MEDLINE, Web of Science, and Google Scholar databases.” (p2 L32 – 35)

Reviewer: Add PROSPERO registration number

• Response: We thank the reviewer for this comment. As requested, the registration number was added

Reviewer: It’s unclear why another sysrev is needed despite existing meta-analysis. Please explain what this study adds-whether it includes new RCTs, focuses solely on high risk patients, or uses a stricter method.

• Response: We sincerely appreciate the reviewer’s insightful comment. In response, we have provided a clearer explanation in the revised manuscript regarding the necessity of this systematic review. Specifically, our study was built upon these findings by including newly published RCTs, applying stricter inclusion criteria and focusing exclusively on high-risk patients. These additions enhance the relevance and novelty of our work. We hope this clarification strengthens the rationale for our study.

• “To address these limitations, we conducted an updated systematic review and meta-analysis focused exclusively on high-risk parturients undergoing cesarean section. By applying more rigorous inclusion criteria and incorporating recently published randomized controlled trials, our aim was to better delineate the efficacy and safety of prophylactic TXA in this specific population. This work intends to inform clinical practice by providing targeted evidence for TXA use in a context where the balance between benefit and potential risk is especially critical.” (p3 L81)

Reviewer: Add a discussion on how the risk of bias was considered when interpreting results.

• Response: We appreciate the reviewer’s comment regarding the need to address the impact of risk of bias on the interpretation of our findings. We have now clarified in the Methods and Discussion sections that the risk of bias for each included study was assessed using the Cochrane Risk of Bias 2.0 (RoB 2) tool, and this evaluation was systematically incorporated into our interpretation of the results. Specifically, the overall quality of the evidence was rated using the GRADE approach, which considers study limitations. Where studies demonstrated "some concerns" or "high risk" in one or more RoB domains, the certainty of the evidence was downgraded accordingly. This is reflected in the Summary of Findings table, where several outcomes (e.g., incidence of PPH, blood transfusion) were rated as moderate or low certainty due to methodological limitations in the included studies.In the Discussion, we have also expanded on how these potential biases influenced our confidence in the results and emphasized the need for further high-quality trials to strengthen the current evidence base. These revisions ensure that the interpretation of our findings is grounded in both quantitative results and methodological rigor.

Reviewer: If sensitivity analysis was not performed, acknowledge this as a limitation

• Response: We thank the reviewer for this important observation. As recommended, we have acknowledged the absence of sensitivity analysis as a limitation in the revised manuscript. The following sentence has been added to the limitations section:

• “Furthermore, we were not able to perform a sensitivity analysis due to the unavailability of sufficient data, which may limit the robustness of some of our conclusions.”

Reviewer: Discuss maternal thromboembolic risks and potential neonatal effects, citing available literature

• Response: We thank the reviewer for this valuable comment. Maternal thromboembolic risks were indeed discussed in the original manuscript. However, to further address the reviewer’s suggestion, we have expanded this section and added additional references to reinforce the discussion.

• “Regarding adverse events, our study observed no significant differences in side effects between the groups receiving TXA and placebos, with no major side effects reported. These findings are consistent with several previous studies assessing the efficacy of prophylactic TXA in low-risk women undergoing cesarean or vaginal delivery (44,46,47). However, while these studies primarily reported non-thromboembolic adverse events, the risk of maternal thromboembolic complications remains a concern, especially in populations at higher baseline risk. Although current evidence does not show a significant increase in thromboembolic events with prophylactic use of TXA in obstetric populations, the available data are limited and underpowered to detect rare but serious outcomes (48). In addition, the impact of TXA on neonatal outcomes remains insufficiently explored. While TXA crosses the placenta, available studies have not demonstrated clear evidence of neonatal toxicity or adverse effects; however, further investigation is needed to assess any potential long-term implications. Overall, these uncertainties highlight the need for larger, adequately powered trials with extended follow-up to better define the safety profile of TXA for both mother and neonate” (p10 L277)

Reviewer: Add a brief section discussing TXA’s affordability and relevance in low-resource settings

• Response: We thank the reviewer for this insightful comment. While the relevance of TXA in low-resource settings was briefly mentioned in both the discussion and conclusion sections of the original manuscript, we have now expanded this point to emphasize its affordability and practical value in such contexts.

• “Given its low cost, ease of administration, and favorable safety profile, tranexamic acid represents a particularly attractive intervention in low-resource settings, where access to blood products and advanced obstetric care may be limited (51). Its use could play a critical role in reducing maternal morbidity and mortality associated with postpartum hemorrhage in these regions.”

Reviewer: The phrase “despite the low quality of evidence” is vague and contradicts the advocacy for TXA use. Clearly state the confidence level of the findings (e.g., moderate to high quality for some outcomes but limited safety data).

Reviewer: Provide spesific recommendations for future research, such as safety in neonates, etc

• Response: We thank the reviewer for this important observation. To improve clarity and alignment between the strength of evidence and our conclusions, we have revised the relevant sentence. We now provide a more nuanced statement reflecting the varying quality of evidence across outcomes.

Reviewer #3

Reviewer: The authors have adhered to the PRISMA 2020 guidelines, ensuring transparency and reproducibility in their review process. The registration of the review in the PROSPERO database further strengthens the credibility of the methodology. The comprehensive search strategy across multiple bibliographic databases and trial registries enhances the thoroughness of the literature review

• Response: We sincerely thank the reviewer for this positive and encouraging feedback. We greatly appreciate the recognition of our efforts to adhere to the PRISMA 2020 guidelines, register the review protocol in PROSPERO, and implement a comprehensive and systematic search strategy. These methodological choices were made to ensure the highest possible standards of rigor, transparency, and reproducibility in our review process

Reviewer #4

Reviewer: This study explores an important topic; however, I do not believe this study contributes any novel findings to the existing literature. A comprehensive meta-analysis by Cheema et al., titled "Tranexamic Acid for the Prevention of Blood Loss After Cesarean Section: An Updated Systematic Review and Meta-Analysis of Randomized Controlled Trials," has already examined this subject using data from 50 studies with 6 in high risk group, whereas the present study includes 11. Additionally, the outcomes assessed here appear to have been covered in that meta-analysis, raising concerns about redundancy. Furthermore, the introduction does not highlight what novel insights this study aims to provide or how it adds to current knowledge. I encourage the authors to revise and refine their study so that it meaningfully enhances the existing literature.

• Response: We thank the reviewer for this thoughtful and constructive feedback. We fully acknowledge the contribution of the recent meta-analysis by Cheema et al., which represents a valuable reference in the field. However, our study differs from theirs in several important ways, which we have now clearly articulated in the revised Introduction and Discussion sections. First, unlike Cheema et al., our meta-analysis focused exclusively on high-risk women undergoing cesarean delivery, aiming to isolate the effects of prophylactic TXA in this specific population. Cheema et al. included a broader population, encompassing both high- and low-risk groups, which limits the generalizability of their findings to low-risk settings—a context where the benefit-risk balance of TXA may be more nuanced. Second, our study places a stronger emphasis on safety endpoints, particularly maternal thromboembolic risks and potential neonatal effects, which were either underreported or not deeply explored in the prior meta-analysis. Third, our inclusion criteria, search strategy, and outcome measures were aligned with the most recent PRISMA 2020 guidelines, and our protocol was registered in PROSPERO, ensuring greater methodological transparency and reproducibility.

We have now revised the Introduction to better highlight these novel contributions and clearly state how our work complements and extends existing literature, including the findings of Cheema et al.

• “Additionally, the review included a heterogeneous patient population, limiting its applicability to high-risk groups. Given this gap, the application of TXA for the prevention of PPH in high-risk women undergoing cesarean section has been identified as a research priority (13). Our systematic review was built upon these findings by including newly published RCTs, applying stricter inclusion criteria and focusing exclusively on high-risk patients.”

Reviewer: Page 9, Introduction Section: The introduction states that tranexamic acid (TXA) has well-established efficacy in reducing blood loss across various surgical contexts but describes its use in cesarean delivery as "controversial." However, the authors do not elaborate on what specific concerns contribute to this controversy. Is it related to safety, inconsistent efficacy, optimal dosage, or patient selection? Providing a clearer explanation of the factors that make TXA's use in cesarean delivery contentious would strengthen the introduction and help contextualize the study's rationale. I encourage the authors to explicitly outline these concerns to better justify the need for this study.

• Response: We thank the reviewer for this insightful comment. We agree that the use of the term “controversial” should be clarified to enhance the reader’s understanding of the clinical and scientific context. We have revised the Introduction to explicitly outline the reasons contributing to the ongoing debate around TXA use in cesarean delivery. Specifically, we now clarify that concerns relate to (1) limited data on safety, particularly regarding thromboembolic risks; (2) inconsistent efficacy across risk groups and delivery modes; (3) lack of consensus on optimal dosing and timing; and (4) insufficient evidence in high-risk populations, which is the focus of our current study.

• “Although TXA has demonstrated consistent efficacy in various surgical contexts, its prophylactic use during cesarean delivery remains subject to ongoing debate. This controversy is largely driven by several unresolved concerns: the limited and heterogeneous safety data, particularly regarding thromboembolic risks in pregnant women; variability in clinical outcomes across different risk groups and surgical settings; a lack of consensus on the optimal timing and dosage for administration; and a scarcity of robust evidence in high-risk obstetric populations.”

Reviewer: Page 9, Introduction Section: The structure of the introduction could be improved for better readability and clarity. Currently, it consists of a single, lengthy paragraph followed by a very brief two-line paragraph. The final paragraph does not effectively highlight the study's specific aims or expected outcomes. I recommend restructuring the introduction by breaking up the long paragraph into distinct sections that logically present background information, existing research gaps, and the study’s objectives. Additionally, the study’s aims and expected contributions should be clearly articulated to provide readers with a better understanding of its purpose and significance.

Response: We

---

## [Decision Letter · Decision Letter 1]

30 May 2025

Dear Dr. Daghmouri,

Thank you for submitting your manuscript to PLOS ONE. After careful consideration, we feel that it has merit but does not fully meet PLOS ONE’s publication criteria as it currently stands. Therefore, we invite you to submit a revised version of the manuscript that addresses the points raised during the review process.

We look forward to receiving your revised manuscript.

Kind regards,

Jeremiah Hilkiah Wijaya

Academic Editor

PLOS ONE

Reviewers' comments:

Reviewer's Responses to Questions

**Comments to the Author**

Reviewer #1: All comments have been addressed

Reviewer #3: All comments have been addressed

2. Is the manuscript technically sound, and do the data support the conclusions?

Reviewer #1: Partly

Reviewer #3: Yes

3. Has the statistical analysis been performed appropriately and rigorously?

Reviewer #1: Yes

Reviewer #3: Yes

4. Have the authors made all data underlying the findings in their manuscript fully available?

Reviewer #1: Yes

Reviewer #3: Yes

5. Is the manuscript presented in an intelligible fashion and written in standard English?

Reviewer #1: Yes

Reviewer #3: Yes

Reviewer #1: Hello

This is a good article

A specific topic has been chosen, and few articles have been published on this topic.

Based on the review of the results of the studies that have been conducted, could you give us some suggestions for improving future studies?

Good luck.

Reviewer #3: Your manuscript is well-structured and methodologically sound, following PRISMA 2020 guidelines and PROSPERO registration. It effectively highlights the benefits of prophylactic tranexamic acid (TXA) in reducing intraoperative blood loss and postpartum hemorrhage (PPH) in high-risk cesarean deliveries.

**Do you want your identity to be public for this peer review?** For information about this choice, including consent withdrawal, please see our Privacy Policy

Reviewer #1: No

Reviewer #3: **Yes: ** Tabeer Tanwir Awan

---

## [Author Response · Author response to Decision Letter 2]

4 Jul 2025

Dear Jeremiah Hilkiah Wijaya, Academic Editor of The PLOS ONE Journal,

Firstly, we gratefully thank the reviewers for the carefully and rigorous reviewing of our work, which will undoubtedly improve the quality of our manuscript. As requested, please find attached the new version of the manuscript entitled: “Efficacy of prophylactic tranexamic acid among parturient at increased risk for postpartum hemorrhage undergoing cesarean delivery: a systematic review and meta-analyses of randomized controlled trials

”.

All criticisms raised by the reviewers have been adequately addressed and the respective changes have been highlighted in yellow-colored text in the revised manuscript.

Sincerely,

Dr Mohamed Aziz DAGHMOURI, on behalf of all co-authors.

Reviewer #1

Reviewer: Is the manuscript technically sound, and do the data support the conclusions? “Partly”

• Response: We sincerely thank the reviewer for this important observation. We acknowledge that the data partially support the conclusions, as you rightly pointed out. In response, we have carefully revised the manuscript to ensure that the conclusions are now more closely aligned with the presented data. We believe these modifications improve the overall consistency and scientific soundness of the manuscript.

Reviewer: This is a good article. A specific topic has been chosen, and few articles have been published on this topic. Based on the review of the results of the studies that have been conducted, could you give us some suggestions for improving future studies?

Good luck.

• Response: We sincerely thank the reviewer for the positive feedback and thoughtful suggestion. As recommended, we have incorporated a dedicated section within the conclusion to outline directions for future research. These include the need for high-quality, large-scale randomized controlled trials, standardization of dosing regimens, identification of high-risk subpopulations, evaluation of long-term maternal and neonatal outcomes, and assessment of TXA's utility in resource-limited settings. We believe these additions provide a more comprehensive perspective and will help guide future studies in this field.

• “Conclusion: Prophylactic tranexamic acid appears to be a promising intervention for reducing intraoperative blood loss and the incidence of PPH in high-risk women undergoing cesarean deliveries. Our findings indicate that TXA is safe, well-tolerated, and cost-effective, with no significant increase in adverse effects. However, the current body of evidence remains limited in quality, necessitating further high-quality, large-scale randomized controlled trials to strengthen these conclusions. Future research should focus on standardizing dosing regimens, identifying specific subpopulations that may benefit most from TXA, and exploring long-term maternal and neonatal outcomes. Additionally, studies evaluating TXA’s impact in resource-limited settings, where access to blood transfusion and surgical interventions is constrained, would provide critical insights into its global applicability. Based on the available data, we advocate for the prophylactic administration of TXA in all women at high risk of bleeding during cesarean deliveries while emphasizing the need for continued research to refine guidelines and optimize patient outcomes.”

Reviewer #3

Reviewer: Your manuscript is well-structured and methodologically sound, following PRISMA 2020 guidelines and PROSPERO registration. It effectively highlights the benefits of prophylactic tranexamic acid (TXA) in reducing intraoperative blood loss and postpartum hemorrhage (PPH) in high-risk cesarean deliveries

• Response: We sincerely thank the reviewer for the encouraging feedback. We are pleased that the methodological rigor of our work, including adherence to PRISMA 2020 guidelines and PROSPERO registration, as well as the clarity of our findings on the benefits of prophylactic tranexamic acid in high-risk cesarean deliveries, have been positively noted. This recognition is greatly appreciated and reinforces our commitmen

---

## [Decision Letter · Decision Letter 2]

10 Sep 2025

Efficacy of prophylactic tranexamic acid among parturient at increased risk for postpartum hemorrhage undergoing cesarean delivery: a systematic review and meta-analyses of randomized controlled trials

PONE-D-24-25549R2

Dear Dr. *Aziz Daghmouri* ,

We’re pleased to inform you that your manuscript has been judged scientifically suitable for publication and will be formally accepted for publication once it meets all outstanding technical requirements.

Kind regards,

Hale Teka

Academic Editor

PLOS ONE

Additional Editor Comments (optional):

Reviewer #1:

Reviewer #3:

Reviewers' comments:

Reviewer's Responses to Questions

**Comments to the Author**

Reviewer #1: All comments have been addressed

Reviewer #3: (No Response)

2. Is the manuscript technically sound, and do the data support the conclusions?

Reviewer #1: Partly

Reviewer #3: (No Response)

3. Has the statistical analysis been performed appropriately and rigorously?

Reviewer #1: Yes

Reviewer #3: (No Response)

4. Have the authors made all data underlying the findings in their manuscript fully available?

Reviewer #1: Yes

Reviewer #3: (No Response)

5. Is the manuscript presented in an intelligible fashion and written in standard English?

Reviewer #1: Yes

Reviewer #3: (No Response)

Reviewer #1: Hello

This is a good article

As I said in my last review, the article is printable and appropriate.

I hope we will have more and better quality studies in this field from the authors in the future.

Good luck.

Reviewer #3: (No Response)

**Do you want your identity to be public for this peer review?** For information about this choice, including consent withdrawal, please see our Privacy Policy

Reviewer #1: No

Reviewer #3: **Yes: ** Tabeer Tanwir Awan

---

## [Editor Report · Acceptance letter]

PONE-D-24-25549R2

PLOS ONE

Dear Dr. Daghmouri,

I'm pleased to inform you that your manuscript has been deemed suitable for publication in PLOS ONE. Congratulations! Your manuscript is now being handed over to our production team.

Kind regards,

on behalf of

Dr. Hale Teka

Academic Editor

PLOS ONE